# Floquet-tailored Rydberg interactions

Luheng Zhao [1], Michael Dao Kang Lee [1], Mohammad Mujahid Aliyu [1] & Huanqian Loh [1,2] ✉

The Rydberg blockade is a key ingredient for entangling atoms in arrays. However, it requires atoms to be spaced well within the blockade radius, which limits the range of local quantum gates. Here we break this constraint using Floquet frequency modulation, with which we demonstrate Rydberg-blockade entanglement beyond the traditional blockade radius and show how the enlarged entanglement range improves qubit connectivity in a neutral atom array. Further, we find that the coherence of entangled states can be extended under Floquet frequency modulation. Finally, we realize Rydberg anti-blockade states for two sodium Rydberg atoms within the blockade radius. Such Rydberg anti-blockade states for atoms at close range enables the robust preparation of strongly-interacting, long-lived Rydberg states, yet their steady-state population cannot be achieved with only the conventional static drive. Our work transforms between the paradigmatic regimes of Rydberg blockade versus anti-blockade and paves the way for realizing more connected, coherent, and tunable neutral atom quantum processors with a single approach.

Ultracold atoms in reconfigurable tweezer arrays have emerged as one of the most powerful and rapidly growing quantum platforms. These systems have demonstrated impressive quantum many-body simulations[1–3], highly stable frequency standards[4,5], and promising quantum computation architectures[6–19].

At the heart of these quantum applications lies entanglement, which is often effected in neutral atom arrays via the Rydberg blockade[20,21]. Among the various entanglement schemes[8–10,22,23], the Rydberg blockade has been widely adopted due to its robustness against position disorder. However, it requires the distance between two atoms to be well within the blockade radius to prevent substantial entanglement infidelity due to blockade error. This constraint reduces quantum gates on Rydberg atoms to a limited range. Improvements in the Rydberg interaction range would increase the qubit connectivity, which could significantly enhance quantum processing efficiency.

Like other quantum processors, neutral atom computations and simulations are limited by decoherence[24]. New methods for extending the lifetime of entangled states would improve the fidelity of quantum gates[7]. However, even with existing levels of decoherence, neutral atom processors continue to demonstrate excellent quantum simulations[1–3]. As these systems continue to develop more robust and efficient ways to initialize quantum states, they are likely to yield more diverse quantum simulation capabilities.

In this work, we report that these neutral atom platforms can be advanced on three critical fronts—extending the blockade-based entanglement range, improving coherence times, and enabling versatile state-preparation schemes—with Floquet frequency modulation (FFM). Our FFM approach is simple and straightforward to implement in all existing neutral atom array experiments. First, we demonstrate that atoms can be entangled outside the traditional blockade radius, as predicted by Refs. 25,26, thereby significantly increasing the useful range of the Rydberg interaction. Furthermore, we show how FFM can protect a two-atom entangled state against Doppler dephasing, which is the typical mechanism limiting entangled-state coherence in a Rydberg atom array. Finally, we propose a robust transfer of closely-spaced atoms from the ground state into an anti-blockaded state. Such a strongly-interacting state cannot be otherwise attained in the steady state with a conventional static drive, yet its realization would open the door to intricate simulations of quantum dynamics.

A versatile approach, FFM has been used to implement efficient random access quantum processors with superconducting circuits[27], generate strongly interacting polaritons with Rydberg atoms in an optical cavity[28], and create exotic states of light[29]. FFM is also analogous

[1]Centre for Quantum Technologies, National University of Singapore, 117543 Singapore, Singapore. [2]Department of Physics, National University of Singapore, 117542 Singapore, Singapore. ✉e-mail: phylohh@nus.edu.sg

to the shaking of optical lattices, which has been used to realize synthetic gauge fields in ultracold atoms[30–32]. In a Rydberg atom array, FFM has been used to stabilize the revivals of quantum many-body scars[33].

In this work, we focus on the entanglement between two atoms (Fig. 1), which serves as a basic ingredient of quantum information processing. In addition, the results obtained here can be generalized to a large atom array by using single-site addressing lasers to select two atoms at a given time.

## Results

### Floquet frequency modulation: model and implementation

When atoms are excited from the ground state $|g\rangle$ to the Rydberg state $|e\rangle$, the resulting dynamics are governed by the Hamiltonian:

$$\frac{H}{\hbar} = -\Delta(t)\sum_{i=1}^{N}\sigma_{ee}^{i} + \frac{\Omega}{2}\sum_{i=1}^{N}\sigma_{x}^{i} + \sum_{i<j}V_{ij}\sigma_{ee}^{i}\sigma_{ee}^{j}, \qquad (1)$$

where $i$ indexes the atom, $V(r) = C_6/r^6$ is the van der Waals interaction between Rydberg atoms, and $\Omega$ is the Rabi frequency. Under FFM, the laser detuning $\Delta(t)$ is modulated sinusoidally in time with modulation amplitude $\delta$ and modulation frequency $\omega_0$ about an offset $\Delta_0$ to give $\Delta(t) = \Delta_0 + \delta\sin(\omega_0 t)$.

For the resonant addressing of an atom-array building block comprising two atoms, we set $\Delta_0 = 0$ and $N = 2$. In the absence of dissipation, the two-atom system can evolve between the $|gg\rangle, |W\rangle = (|ge\rangle + e^{i\phi}|eg\rangle)/\sqrt{2}$, and $|ee\rangle$ states, where $\phi$ denotes the relative phase between the atoms arising from their initial positions. With the application of FFM, the two-atom Hamiltonian can be transformed to a new Hamiltonian in a rotating frame, such that the coupling strengths for the $|gg\rangle \leftrightarrow |W\rangle$ and $|W\rangle \leftrightarrow |ee\rangle$ transitions are respectively rescaled to[25,26]:

$$\Omega_a(t) \propto \Omega\sum_{m=-\infty}^{\infty}J_m(\alpha)e^{im\omega_0 t + im\frac{\pi}{2}}, \qquad (2)$$

$$\Omega_b(t) \propto \Omega\sum_{m=-\infty}^{\infty}J_m(\alpha)e^{i(m\omega_0 + V)t + im\frac{\pi}{2}}, \qquad (3)$$

where $J_m(\alpha)$ is the $m^{th}$ order Bessel function of the first kind with modulation index $\alpha = \delta/\omega_0$.

A high modulation frequency simplifies the picture as only the resonant terms dominate the coupling strength. Explicitly, the Rabi frequency for $|gg\rangle \leftrightarrow |W\rangle$ is dominated by $J_0(\alpha)$, whereas the resonance condition for $|W\rangle \leftrightarrow |ee\rangle$ is met by setting $m\omega_0 = -V$. The interplay of these two resonance conditions dictates the physics behind the results presented in this study.

To implement FFM, we send a Rydberg excitation laser through an acousto-optic modulator driven with a time-varying frequency (Fig. 1a). Since the dynamics of the two-atom system depends critically on the modulation index $\alpha$, we take care to calibrate the modulation amplitude $\delta$, which can differ from the specified amplitude due to the finite modulator bandwidth. We further minimize the residual amplitude modulation arising from the frequency-dependent diffraction efficiency of the acousto-optic modulator (see Supplementary Note 3). We note that FFM can be easily implemented given the typical bandwidth of commercially available acousto-optic modulators.

We point out that FFM is distinct from the Floquet engineering techniques used to control dynamics in Rydberg atoms[34–36]. The latter involves changing the effective Hamiltonian by applying fast, periodic pulses with well-defined phase relations. These pulses are used in the microwave domain to address either two ground states in a Rydberg-dressed system or two Rydberg states. In contrast, FFM directly transforms the effective couplings. Further, FFM does not use periodic pulses, which would have been challenging to implement in the optical domain on the time scale of Rabi oscillations between the ground and Rydberg states.

Our experiment procedure begins with the $D_1$ Λ-enhanced loading of single $^{23}$Na atoms into two optical tweezers[37–39]. We excite the $^{23}$Na atoms from the ground state $|g\rangle = |3S_{1/2}, F = 2, m_F = 2\rangle$ to the Rydberg state $|e\rangle = |59S_{1/2}, m_J = 1/2\rangle$ via the intermediate state $|m\rangle = |3P_{3/2}, F = 3, m_F = 3\rangle$ with two photons at 589 nm and 409 nm. The Rydberg laser intensities and single-photon detuning $\Delta'$ are chosen to give an effective single-atom Rabi frequency of $\Omega/(2\pi) = 1.0$ MHz. FFM is only applied to the 589 nm excitation laser. The optical tweezers are switched off during Rydberg excitation and turned back on at the end

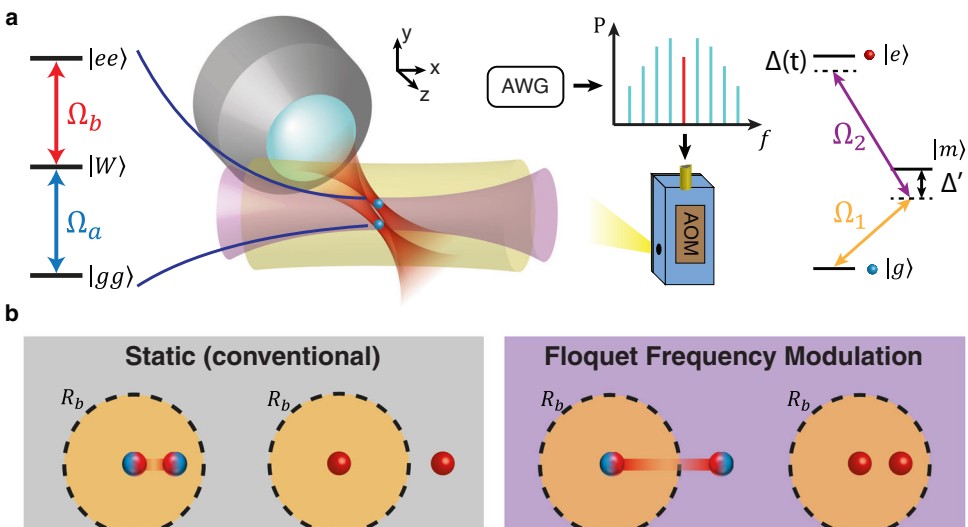

**Fig. 1 | Using Floquet frequency modulation to transform Rydberg interactions. a** Experiment setup for implementing FFM on two tweezer-trapped $^{23}$Na atoms with counter-propagating Rydberg lasers at 589 nm ($\Omega_1$) and 409 nm ($\Omega_2$). The 589 nm Rydberg laser is frequency-modulated with an acousto-optic modulator (AOM) driven by an arbitrary waveform generator (AWG). **b** Transforming between the two regimes of interactions: Rydberg blockade (two entangled atoms depicted as blue/red spheres) versus anti-blockade (two red atoms). With the conventional static drive, the Rydberg blockade regime can only be accessed with the atoms positioned well within the blockade radius $R_b$, while atoms spaced farther than the blockade radius experience anti-blockade. The counter-intuitive regime of interactions can be exploited with Floquet frequency modulation, where atoms outside the static blockade radius experience Rydberg blockade and vice versa, as predicted by refs. 25,26.

of the excitation sequence to image the atoms. Using $D_1$ imaging, the ground (Rydberg) state is detected as the presence (absence) of a recaptured atom. The false positive error (3%) is dominated by the imaging survival probability, whereas the false negative error due to the decay of Rydberg atoms is estimated to be 3%.

## Extended Rydberg blockade range

For a static Rydberg excitation ($\alpha = 0$), suppression of population in the $|ee\rangle$ state is an important prerequisite for the high-fidelity generation of the entangled $|W\rangle$ state. Therefore, quantum gates are typically carried out between atoms spaced well within the Rydberg blockade radius $R_b$, where $V(R_b) = \Omega$. Under FFM, the two-atom couplings are rescaled by Bessel functions (Eqs. (2) and (3)), giving an intuitive picture for the transformation of the Rydberg blockade radius.

As an example, we consider two atoms spaced farther than the static blockade radius, such that their interaction strength is given by $V(r) = 0.5\,\Omega$. Figure 2a and b show the calculated time-averaged $|ee\rangle$ populations and the maximum $|W\rangle$ fidelities, respectively, as a function of the modulation frequency and modulation index. In the regime of high modulation frequency ($\omega_0 \geq 2\,\Omega$), the dynamics are relatively robust and are dictated by the Bessel functions. At the Bessel function zeros ($J_0(\alpha) = 0$), the $|ee\rangle$ population is the most suppressed. However, it would be misleading to use these Bessel function zeros for generating entangled states, as the population is actually trapped in its initial $|gg\rangle$ state. To generate entangled states, it is optimal to select $\alpha$ slightly away from the Bessel function zeros. Here, the rescaled Rabi frequency $\Omega_a \propto J_0(\alpha)\Omega$ remains finite but small compared to the interaction, such that errors arising from populating the $|ee\rangle$ state can still be suppressed while the $|W\rangle$ state can be generated with good fidelity. Furthermore, by operating near a higher order Bessel function zero, the $|W\rangle$ state can be realized with high fidelity over a wider range of modulation indices. Given the finite modulation bandwidth of the acousto-optic modulator, it is advantageous to work with a modulation frequency that is large enough compared to the Rabi frequency, but still low enough to allow a high modulation index (e.g. $\omega_0 = 3\,\Omega$).

Figure 2c shows the measured blockade enhancement under FFM, taken with $\omega_0 = 3\,\Omega$ and at various modulation indices. Compared to the static excitation, the dynamics of the $|ee\rangle$ state under FFM are significantly suppressed (inset, $V = 0.8\,\Omega$). The observed Rydberg blockade over a range of atom spacings ($r/R_b = 0.93 - 1.3$, inferred from the normalized interaction range $V/\Omega = 0.22 - 1.5$) agrees well with the theory

simulations. Through an appropriate choice of $\alpha$, we observe either population trapping (Fig. 2d, $\alpha = 5.5$) or coherent dynamics between the $|gg\rangle$ and $|W\rangle$ states (Fig. 2e, $\alpha = 6.9$). The $|W\rangle$ fidelity achieved in Fig. 2e is determined from Monte Carlo simulations to be 0.77(5). The observed fidelity is primarily limited by the coherence of the Rydberg excitation lasers, which can be improved with cavity filtering techniques[40].

Where the coherence of the Rydberg excitation laser and the finite lifetime of the Rydberg state are no longer dominant constraints, the entangled state fidelity can be optimized by choosing $\alpha$ in exchange for longer gate times. For instance, one can use FFM to access a $|W\rangle$ fidelity of 0.98 using $\alpha = 11.1$ even at a small interaction strength of $V = 0.5\,\Omega$. To access the same $|W\rangle$ fidelity with the static Rydberg excitation scheme, the atoms would have had to experience an interaction strength of $V = 4.9\,\Omega$, effectively extending the Rydberg blockade range by a factor of $(4.9/0.5)^{1/6} > \sqrt{2}$ (see Supplementary Note 6). In other words, for an atom array with a fixed square geometry, the static scheme would have allowed only the four nearest neighbors to be entangled in a pairwise manner with the center atom, whereas implementing FFM would allow the next-nearest neighbors on the diagonals to be pairwise entangled with the center atom, thereby potentially doubling the number of qubit connections on demand.

In the rest of this paper, we switch to working well within the Rydberg blockade radius and demonstrate two more useful features of the FFM.

## Protection of entangled state against dephasing

The $|W\rangle$ state can be dynamically stabilized[26] under the same conditions that give rise to population trapping ($J_0(\alpha) = 0$). We demonstrate this by first transferring atoms to the $|W\rangle$ state with a static, resonant $\pi$-pulse in the Rydberg blockade regime ($V = 8\,\Omega$). Subsequently, we apply the FFM ($\omega_0 = 6\,\Omega$, $\alpha = 5.5$) for 2 μs, before returning to the static drive. The Rabi frequency is kept constant throughout the sequence (Fig. 3a). During the FFM, the dynamics of the $|W\rangle$ state are frozen (Fig. 3b), in contrast to the case where the static drive is applied throughout the sequence (Fig. 3c).

Instead of FFM, one can also trivially keep atoms in the $|W\rangle$ state by turning off the excitation lasers after the first $\pi$-pulse. We refer to this alternative as the laser-free scheme. In each case (laser-free versus FFM), the entangled state coherence is limited by relative Doppler shifts between the two atoms[40], and can be measured by first applying a $\pi$-pulse to drive $|gg\rangle$ to $|W\rangle$, then waiting for a variable time

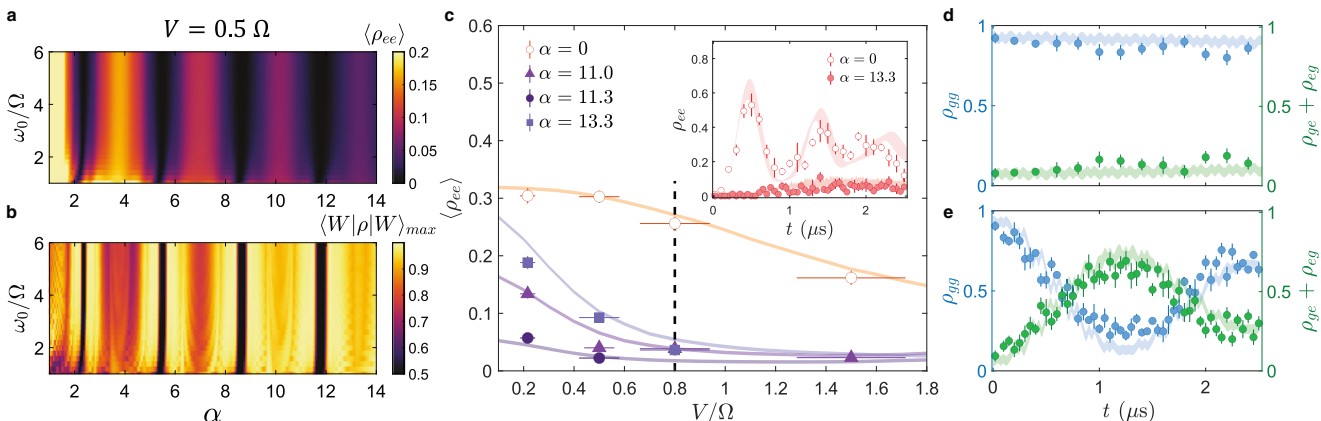

**Fig. 2 | Extended Rydberg blockade range. a** Calculated $|ee\rangle$ population, time averaged over 5 μs, and **b** calculated maximum $|W\rangle$ fidelities as a function of normalized modulation frequency $\omega_0/\Omega$ and modulation index $\alpha$. **c** Measured $|ee\rangle$ population, time-averaged over 2.5 μs, as a function of normalized interaction strength $V/\Omega$ for $\omega_0 = 3\,\Omega$ and different modulation indices $\{\alpha\}$. Horizontal error bars are attributed to the position uncertainty of the atoms. Solid lines indicate the numerical modeling results for the corresponding $\alpha$. (Inset) Population dynamics

$\rho_{ee}$ at $V = 0.8\,\Omega$. **d, e** Population dynamics for (left axis, blue) $\rho_{gg}$ and (right axis, green) $\rho_{eg} + \rho_{ge}$, measured at $V = 0.8\,\Omega$ and $\omega_0 = 3\,\Omega$. The shaded curves reflect the results of Monte Carlo simulations. **d** Population trapping in $|gg\rangle$ is observed for $\alpha = 5.5$, where $J_0(\alpha) \approx 0$. **e** At $\alpha = 6.9$, the $|W\rangle$ state is generated with a calculated maximum fidelity of 0.77(5). In panels **c**–**e**, the displayed vertical error bar associated with each data point reflects the 1σ confidence interval.

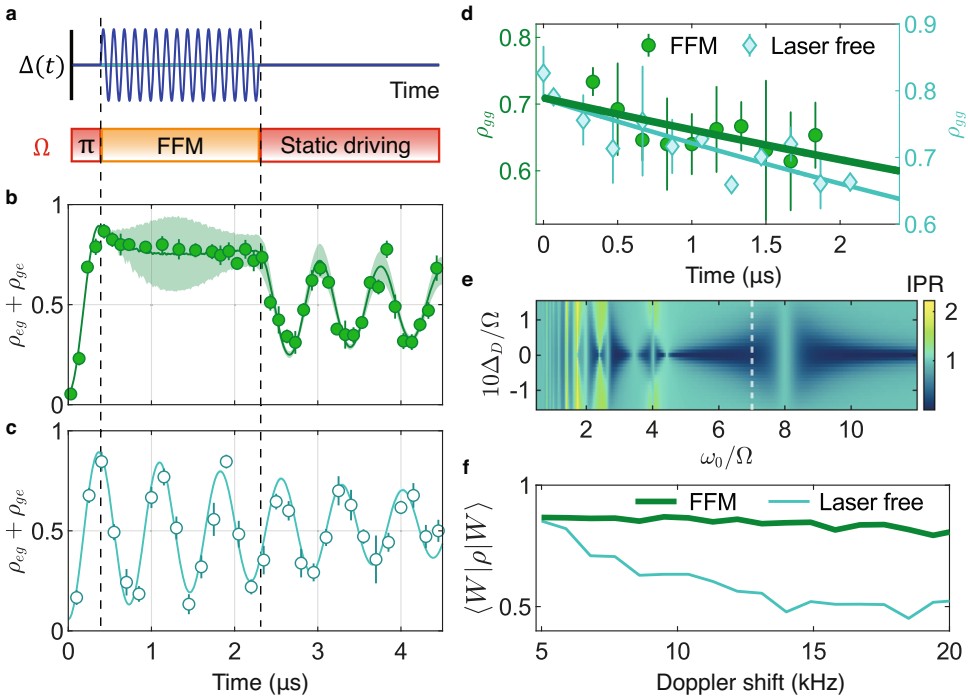

**Fig. 3 | Dynamical stabilization of entangled states at the Bessel function zero $\alpha = 5.5$ and at $V = 8\,\Omega$. a** Pulse sequence comprising an initial $\pi$ pulse, 2 $\mu$s of FFM ($\omega_0 = 6\,\Omega$), followed by static driving. **b, c** Observed population dynamics $\rho_{eg} + \rho_{ge}$ **b** under the above FFM pulse sequence versus **c** under a continuous, resonant static drive. The solid lines account for the decoherence effects as modeled by Lindblad superoperators. The shaded bands depict Monte Carlo simulations that include the atom position uncertainty. **d** Comparison of $|W\rangle$ decay for (left axis, green) FFM versus (right axis, blue) a laser-free evolution. The FFM

evolution yields a decay time of 14(5) $\mu$s whereas the laser-free evolution yields a decay time of 11(2) $\mu$s. Error bars in panels (**b**–**d**) depict the $1\sigma$ confidence interval. **e** Calculated IPR as a function of rescaled Doppler shift $10\Delta_D/\Omega$ and normalized modulation frequency $\omega_0/\Omega$. (White dashed line) At $\omega_0 = 7\,\Omega$, the IPR is approximately 0 over an extended range of Doppler shifts, which is desired for robust dynamical stabilization of $|W\rangle$. **f** Calculated $|W\rangle$ fidelity for different Doppler shift widths after 20 $\mu$s of (green) FFM or (blue) laser-free evolution. Here the FFM parameters are $\omega_0 = 7\,\Omega$, $\alpha = 5.5$.

$0 \le t \le 2\,\mu$s before applying another $\pi$-pulse to measure the population in $|gg\rangle$. Figure 3d compares the measured decay of the $|W\rangle$ state for both cases, where the FFM sequence yields a fit decay time of 14(5) $\mu$s and the laser-free scheme shows a comparable decay time of 11(2) $\mu$s.

With a judicious choice of parameters, FFM can protect the entangled $|W\rangle$ state from dephasing and maintain its coherence over the laser-free case. Intuitively, the protection arises from a high-frequency interruption of the dephasing process. More specifically, although we null out the Rabi coupling between $|gg\rangle$ and $|W\rangle$ with FFM, the off-resonant coupling between $|W\rangle$ and $|ee\rangle$ remains. Since the $|ee\rangle$ state is robust against drifts in relative phase shifts between the atoms, the Doppler dephasing is periodically interrupted whenever the system evolves to pick up the small $|ee\rangle$ admixture. The fast residual oscillation is washed out in a time average, giving rise to a net protection of the $|W\rangle$ state.

To access the entanglement-protection regime, one needs to ensure that the $|W\rangle$ state has a strong overlap with an eigenstate of the Floquet Hamiltonian $|\phi_k(0)\rangle$. The overlap $p_k = |\langle \phi_k(0)|W\rangle|^2$ is parameterized by the inverse participation ratio (IPR), which is defined[26] to be $\Pi^{|W\rangle} = (1/\sum_k p_k^2) - 1$. A low IPR is preferred for robust dynamical stabilization of the $|W\rangle$ state. Operating, for instance, at $\omega_0 = 7\,\Omega$ yields a low IPR over an extended range of Doppler shifts (Fig. 3e). After 20 $\mu$s of FFM application, the $|W\rangle$ state fidelity under FFM is predicted to be significantly higher than that for the laser-free case (Fig. 3f). The higher fidelity under FFM is maintained over a large range of Doppler shifts, demonstrating the robustness of entanglement under FFM.

### Enhanced Rydberg anti-blockade dynamics
We now turn our attention to Rydberg anti-blockade states[41], which are promising for quantum simulations of interesting dynamics such as

that of epidemics[42] and of flat band systems in condensed matter physics[43]. Rydberg anti-blockade (i.e. $|ee\rangle$ population) is typically achieved for two atoms spaced outside the blockade radius[23]. Within the blockade radius, Rabi oscillations between the $|gg\rangle$ and $|ee\rangle$ states can still be realized[44] with a finite static detuning, e.g. $\Delta_0 = V/2$. However, this comes at the expense of slower dynamics, particularly when $V \gg \Omega$, as the effective Rabi frequency is set by $\Omega^2/\Delta_0$.

On the other hand, FFM provides a convenient handle to access the $|ee\rangle$ state from the $|gg\rangle$ state by fulfilling both resonance conditions[25,26] described in Eqs. (2) and (3) (Fig. 4a). For instance, setting $\omega_0 = V$ causes the coupling strength for the $|W\rangle \leftrightarrow |ee\rangle$ transition to be dominated by $J_1(\alpha)$, while $J_0(\alpha)$ continues to dominate the $|gg\rangle \leftrightarrow |W\rangle$ coupling strength. Consequently, choosing a modulation index that gives a large value for both $J_1(\alpha)$ and $J_0(\alpha)$, such as $\alpha = 1.4$, realizes the Rydberg anti-blockade (Fig. 4b). Figure 4c shows the corresponding two-atom population dynamics for $V = \omega_0 = 6\,\Omega$ and $\alpha = 1.4$, where the $|ee\rangle$ state can be accessed with a speedup over the off-resonant static drive. The observed $|ee\rangle$ population is limited by the finite position spread of the atoms at about 1 $\mu$K, which gives rise to a range of interaction strengths, over which the resonance condition $\omega_0 = V$ does not always hold. This problem can be mitigated by optimal cooling techniques, such as motional ground-state cooling[45], and by increasing the atom-laser coherence (see Supplementary Note 4).

To boost the $|ee\rangle$ population further, we propose to combine FFM with stimulated Raman adiabatic passage (STIRAP). A reliable state-preparation method, STIRAP has been used to initialize atoms in multiply-excited Rydberg states for studies of symmetry-protected topological phases[46], spin transport in one-dimensional systems[36], and more. However, to date, such STIRAP transfer has only been demonstrated on atoms spaced outside the blockade radius. Reducing the

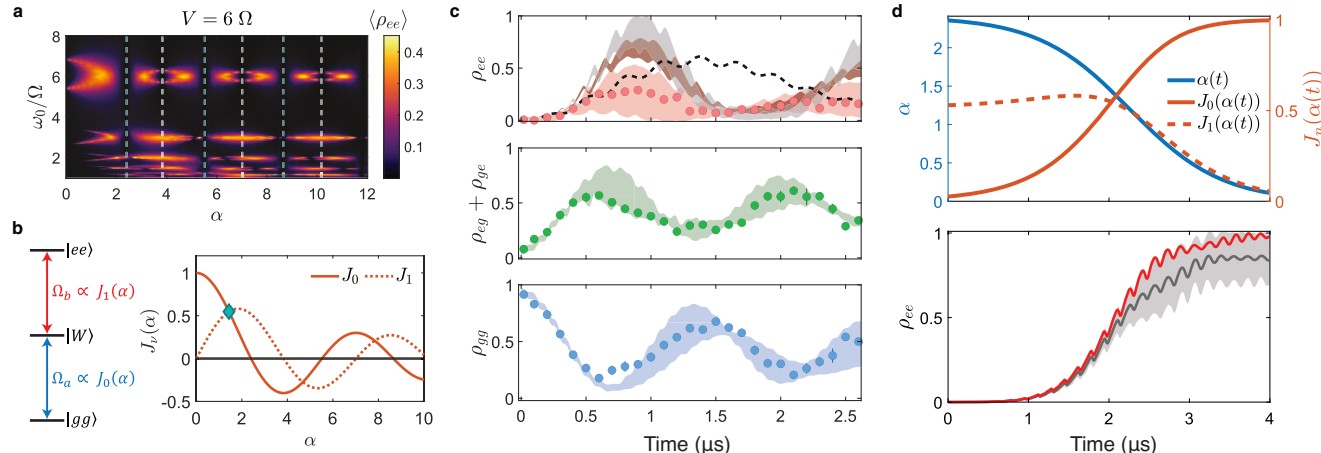

**Fig. 4 | Enhanced Rydberg anti-blockade dynamics at $V = \omega_0 = 6\,\Omega$. a** Calculated $|ee\rangle$ population, time-averaged over 10 μs, as a function of normalized modulation frequency $\omega_0/\Omega$ and modulation index $\alpha$. The green and white dashed lines indicate positions at which $J_0(\alpha) = 0$ and $J_1(\alpha) = 0$, respectively. **b** The two-atom Rabi frequencies $\{\Omega_a, \Omega_b\}$ depend strongly on the modulation index. (Green diamond) At $\alpha = 1.4$, both Rabi frequencies have large values and can be used to access Rydberg anti-blockade states. **c** Measured population dynamics (top) $\rho_{ee}$, (middle) $\rho_{eg} + \rho_{ge}$, and (bottom) $\rho_{gg}$ for $\alpha = 1.4$, which are in good agreement with Monte Carlo simulations represented by shaded curves (in same color as data points). Error bars represent the $1\sigma$ confidence interval. (Brown shading) The FFM-induced anti-blockade population can be further increased with the help of ground-state cooling. The gray shading indicates the simulated $\rho_{ee}$ achieved by FFM under both ground-state cooling and enhanced coherence times of 74 μs. In general, the FFM dynamics are faster than that with an off-resonant static drive (black dashed line), where $\Delta_0 = V/2$. **d** Simulation of steady-state Rydberg anti-blockade, achieved by combining FFM with STIRAP. (Top) The modulation index $\alpha$ is smoothly varied from 2.4 to 0, such that the two-atom Rabi frequencies change in time despite holding $\Omega$ constant. (Bottom) Calculated population dynamics $\rho_{ee}$ for the proposed STIRAP sequence. (Red line) In the absence of imperfections, $\rho_{ee}$ can be as high as 0.98. (Gray line) With ground state cooling and enhanced coherence times of 74 μs, the mean steady-state population becomes $\rho_{ee} = 0.85$, with one standard deviation of the Monte Carlo sample distribution depicted as the half width of the light gray band.

spacing between atoms would be desired for stronger interactions, yet the steady-state population of multiply-excited Rydberg states cannot be achieved by applying STIRAP with the conventional static excitation scheme.

On the other hand, FFM offers a straightforward path for populating the $|ee\rangle$ state through STIRAP for atoms well within the blockade radius. The modulation index is a flexible degree of freedom that controls both two-atom coupling strengths simultaneously. The adiabatic transfer of atoms from the initial $|gg\rangle$ state to the final $|ee\rangle$ state can be accomplished by ramping the modulation index from its first Bessel function zero ($\alpha = 2.4$; $J_0(\alpha) = 0$) down to $\alpha = 0$ (where $J_1(\alpha) = 0$) over time (Fig. 4d). The adiabatic ramp needs to be performed quickly compared to the decoherence of the $|W\rangle$ state but slowly compared to the coupling strengths $\{\Omega_a, \Omega_b\}$. We note that the effective Rabi frequency for each transition varies asymmetrically in time despite the laser intensities being kept constant throughout the transfer.

## Discussion

We have demonstrated that FFM is a versatile approach that can be used to increase the entanglement range, protect the entangled state coherence, and initialize strongly-interacting states in Rydberg atom arrays. The full advantages afforded by FFM can be realized by working with a Rydberg state of higher principal quantum number so as to access a longer lifetime and by making several technical upgrades. These include improving the Rydberg laser coherence though cavity filtering[40], suppressing off-resonant scattering rates by increasing the detuning from the intermediate state[24,47], and reducing the position disorder through ground-state cooling[45], all of which have already been demonstrated separately in other experiment setups.

Our results can be extended to the generation and protection of entanglement between two long-lived ground states, where the entanglement is mediated by excitation to a Rydberg state[9]. For an arbitrary graph depicting a particular geometric arrangement of single atoms at its vertices, FFM enables connectivity between any two atoms through an appropriate choice of the modulation index. FFM can also

be combined with mobile optical tweezers that can transport entanglement over longer distances[11]. In such a combination, FFM can reduce both the duration and the number of moves required to perform pairwise entanglement across the entire array, thereby leading to a more streamlined quantum information processing architecture[7]. We further note that the enlarged blockade range can be used to entangle multiple atoms simultaneously (see Supplementary Note 7). In this case, FFM can effect the dynamical control of the Rydberg blockade range beyond that accomplished by simply tuning the Rydberg laser intensity, thereby offering a flexible way to access quench dynamics in quantum many-body simulations[1,2].

Importantly, our work redefines the ability to access the two paradigmatic regimes of interactions—Rydberg blockade versus anti-blockade, which have thus far been mostly governed by the precise positioning of atoms and the static blockade radius. The FFM not only enhances the programmability of Rydberg atom arrays but also enables steady-state Rydberg anti-blockade for atoms spaced within the blockade radius, which cannot be otherwise attained with conventional static schemes. These results open the door to realizing arrays of closely spaced Rydberg atoms, including those in long-lived circular Rydberg states that are attractive for quantum computing and simulation[48–50].

## Methods

Our experiment begins with loading individual $^{23}$Na atoms into a pair of magic-wavelength tweezer sites from a precooled reservoir[37]. Each of the two traps is formed by focusing 615.87 nm light to a beam waist of 0.78 μm and is set to a trap depth of 1.3 mK during the $D_1$ Λ-enhanced loading. We then image once with Λ-enhanced $D_1$ imaging to postselect experiment shots in which both traps are loaded. Unlike in[37], here the counterpropagating pair of $D_1$ imaging beams are sent along the $y$ axis and are ($\sigma^+, \sigma^-$) polarized with a total intensity of 6 $I_{\text{sat}}$ ($I_{\text{sat}} = 6.26$ mW/cm$^2$) and $I_{\text{repump}}/I_{\text{cool}} = 0.11$.

We initialize the atoms in the ground state $|g\rangle = \left|3S_{1/2}, F = 2, m_F = 2\right\rangle$ with 99.1% fidelity by optically pumping the

atoms in the presence of a bias magnetic field of 2.7 G along the $x$ axis. The trap depths are then ramped linearly down to 7 μK in 0.8 ms, yielding an atom temperature of about 1 μK as determined by comparing the measured release-and-recapture probability against Monte Carlo simulations. The tweezer traps are subsequently turned off during Rydberg excitation.

To drive the atoms to the $\left|59S_{1/2}, m_J = 1/2\right\rangle$ Rydberg state, a global two-photon excitation pulse consisting of a $\sigma^+$-polarized 589 nm beam and a counter-propagating $\sigma^-$-polarized 409 nm beam is sent to the atoms along the $x$ axis. We realize single-photon Rabi frequencies of $\{\Omega_1, \Omega_2\} = 2\pi \times \{70, 26\}$ MHz and single-photon detunings of $\Delta' \approx 2\pi \times 850$ MHz, red-detuned from the $\left|3P_{3/2}, F = 3, m_F = 3\right\rangle$ intermediate state. We note that the atom array axis (along $y$) being orthogonal to the direction of Rydberg laser propagation (along $x$) is suboptimal, but this is limited in our experiment setup by the residual coma aberration of the microscope objective, and can be easily resolved with an improved objective. Nevertheless, we equalize the Rabi frequencies for the two atoms by realigning the 409 nm laser with a piezoelectric mirror from time to time. The combined two-photon Rydberg Rabi frequency is typically $\Omega = 2\pi \times 1.0$ MHz. Our atom array orientation also means that the relative phase $\phi$ in the expression for $|W\rangle$ is close to zero.

After the Rydberg excitation pulse, atoms in the ground (Rydberg) state will be recaptured (repelled) by the tweezer trap (restored trap depth of 1.35 mK). Finally, the $D_1$ imaging beams are applied a second time to read out the population of atoms in the ground versus Rydberg state. A bright (dark) image nominally indicates the presence of an atom in its ground state (Rydberg state). The data reported in the main text include the state preparation and measurement errors (see Supplementary Note 1). Each data point is the average of at least 200 experiment cycles.

## Data availability

The data that support the findings of this study are available on https://doi.org/10.5281/zenodo.8347276.

## Code availability

The simulation codes used in this work are available from the corresponding author upon request.

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

## Acknowledgements

We acknowledge Travis Nicholson and Wen Wei Ho for stimulating discussions, as well as Krishna Chaitanya Yellapragada for technical assistance with the experiment setup. This research is supported by the National Research Foundation, Singapore and A*STAR under its Quantum Engineering Programme (NRF2021-QEP2-02-P09) and its CQT bridging grant.

## Author contributions

H.L. initiated the experimental study. L.Z., M.D.K.L. and M.M.A. carried out the experiments. L.Z., M.D.K.L. and H.L. performed the theoretical modeling. All authors contributed to the discussion of results and manuscript preparation.

## Competing interests

The authors declare no competing interests.
