## [Peer Review File · Nature Communications]

Floquet-Tailored Rydberg InteractionsREVIEWER COMMENTS

Reviewer #1 (Remarks to the Author):

L. Zhao et al., report on the control of Rydberg blockade effects via Floquet frequency modulation method for atoms in optical tweezer arrays. A Rydberg state of a Na atom is created via the two-photon excitation from the ground state in two-dimensional optical tweezer arrays. In the standard scheme of static Rydberg excitation, a Rydberg blockade mechanism prohibits the formation of two Rydberg state for two atoms within the Blockade radius, determined by the interatomic interaction between the two Rydberg states and the two-photon Rabi frequency. In this work, by developing a dynamic, or Floquet frequency modulation method, the authors demonstrated three major results: 1) extension of the Rydberg blockade radius, 2) protection of entangled W-state, and 3) formation of anti-blockade state. An efficient formation of anti-blockade state via STIRAP is also proposed and discussed. The observed results can basically be qualitatively understood via effective control of Rabi frequency. The supplementary information provides the details of the underlying physics and the above experiments.

I think this paper interesting. The technique of Floquet frequency modulation is clearly demonstrated, providing a new tool for quantum simulation and quantum computation using neutral atom tweezer arrays. Therefore, I am positive to recommend this paper for publication for Nature Communication. However, I have several comments and questions for consideration by the authors. The authors' answers and response will benefit broader range of readers.

1) Concerning Extended Rydberg blockade range.

I believe that working with static, small Rabi frequency, instead of modulation, can in principle, gives the same result. It is nice that the authors discuss the merit of FFM scheme over the static weak excitation scheme. Note that, in this discussion, the authors should compare the two schemes on the SAME Rabi frequencies, meaning that in the FFM scheme the authors should use the suppressed Rabi frequency $J_0 \times \Omega$, instead of bare Ω as in the comparison in SI IV.

In addition, smaller effective Rabi frequency requires the longer time for Rydberg two-qubit gate, leading to larger infidelity of the two-qubit gate, due to the finite lifetime of Rydberg state. Can the authors discuss this point ?

2) Concerning Protection of entangled state against dephasing

Intuitive explanation on the stronger protection of the entangled state via FFM over the free evolution is only briefly provided as arising from a high-frequency interruption of the dephasing process. I think this is not enough and more detailed discussion is helpful for broader range of readers.

In particular, I am wondering if this protection is related with a quantum Zeno effect.

3) Concerning Enhanced Rydberg anti-blockade dynamics

I believe that the Rydberg excitation with static, two frequency components, one resonant with the gg to W and another with W to ee , instead of modulation, can in principle, give the same result. Actually, the authors used this later scheme for the measurements described in Supplementary Information II. It is nice that the authors discuss the merit of FFM scheme over the static, two frequency components, excitation scheme.

Since realizing the STIRAP scheme that the authors discussed is rather straightforward for the authors, I am wondering if there is any technical issue. If so, it is nice to discuss this point.

4) Concerning Fig. 4

Fig.4c): I cannot find any explanation for red, green, and blue shading. I can imagine they are MC simulations, but explicit explanations are helpful (although in SI, there are related sentences.)

Fig. 4d): Similarly, I cannot find any explanation for grey shading.

5) Concerning Methods

At the beginning, the authors state "...two magic-wavelength tweezers ...". I am a little bit confused about this wording. Does this "two" mean "two sites"? How many two-sites are used in the experiment?

6) Concerning Supplementary Information I

In the evaluation of false negative error, the authors used the BBR-limited lifetime for the Rydberg state. Since the conversion to the ground state should occur for this error and the BBR induces the transfer to other Rydberg states, this BBR-limited lifetime does not seem to provide the relevant time constant.

7) Concerning Supplementary Information II

Can the authors compare their method with much simpler one from the imaging magnification? The presented method looks nice, but could be model-dependent?

8) Concerning Supplementary Information Extended Data Fig. 5

I am a little bit confused. In the caption, are the green and grey swapped?

In addition, it may be helpful to explicitly show that the displayed curves are for the static drive scheme (shown possibly in the figure).

Reviewer #2 (Remarks to the Author):

The paper discusses the controlled engineering of the quantum state of two Rydberg qubits via periodic modulation of the frequency of the light field. The authors successfully demonstrated various exciting phenomena, which is an excellent achievement. The results are relevant to the present scenario of achieving quantum technological applications using neutral atoms, particularly the Rydberg atoms. For instance, in Fig. 2, the population trapping in the gg state and the blockade dynamics at small interactions (well below the blockade threshold). They also discussed how to improve the fidelity of the W state for small interaction strengths.

In Fig.3, the authors demonstrated the freezing of the population in the entangled W state. The lifetime of the same state under FFM is compared with that of the laser-free case, which clearly shows that periodic driving enhances the lifetime. The same figure demonstrates the robustness of the entanglement stabilization by FFM against Doppler shifts.

The antiblockade at sufficiently large Rydberg-Rydberg interactions are discussed in the final part of the draft. A novel method combining the FFM with STIRAP is proposed to enhance the ee -state population. Several future perspectives are included.

The results are very solid, and also the paper is well written. I recommend the paper to be published in nature communications after incorporating the changes prescribed below.

Specific comments

The introduction's second paragraph states, "...to prevent substantial entanglement infidelity due to double Rydberg excitations". In a general context, the statement is not accurate. I feel, for instance, one may need a double excitation population to realize a maximally entangled state like $|gg\rangle + |ee\rangle$. So, a better reformulation would be suitable.

In the paragraph above eq.2, the state $|g\rangle + e^{i\phi}|g\rangle$, does the ϕ can take any values, for instance, π ? for which one creates the singlet state $|g\rangle - |g\rangle$? Or ϕ is restricted to some values?

The discussions on the second page, starting from the paragraph, "We point out that FFM is distinct from the Floquet engineering ..." and the paragraph following that, are best suited for the introduction.

The references Refs.[25, 26] above eqs.(2) and (3) discuss theoretically some of the main ideas in this paper, such as blockade at small interactions, antiblockade at large interactions, and Bell-states freezing using FFM. That needs to be appropriately credited in the draft and best suited to be included in the introduction. As the present paper demonstrates the experimental results, it has its merit.

It would be useful if authors can provide a reference for the statement “the protection arises from a high frequency interruption of the dephasing process”. As it is very important point in the scenario of the experiment. Or add more insights or explanations into it, how the FFM can suppress the decoherence.

Reviewer #3 (Remarks to the Author):

The manuscript reports the experimental realization of Floquet frequency modulation (FFM) in Rydberg atom pairs. The demonstrated FFM adds a new component to the Rydberg quantum processing toolbox. Most importantly, the authors show that FFM allows the suppression of double Rydberg excitation beyond the conventional blockade radius, and thus can be used to entangle atoms that are further separated. This new technique can in principle be used to extend the quantum connectivity in a Rydberg atom array. The authors also apply FFM in the protection of entangled $|W\rangle$ state and in the preparation of double Rydberg excitations. The paper is soundly written in general, and the claims are well supported with experimental and numerical results.

I think this work is novel and will be of interest to the growing community of Rydberg-array-based quantum physics. Therefore, I would be happy to recommend its publication in Nature Communications, if my following comments are well addressed.

1. The authors demonstrate and discuss the application of FFM in entanglement generation, but can it also be used in a Rydberg quantum logic gate? If yes, what are the Pros and Cons vs the conventional blockade-based Rydberg gate scheme?

2. The finite anti-blockade population is attributed to the interaction disorder from the position spread of the atoms with 1 μ K temperature, and the authors propose to mitigate this effect with further cooling. It seems to me that, the severe trap-depth ramping from 1.3 mK to 7 μ K would largely increase the initial atomic wavepacket distribution. So, if the interaction disorder is mainly caused by initial position spread (instead of initial atomic velocity), then maybe it can be mitigated just by going easy on the trap-depth ramping?

Response to the referees

We thank all three referees for their careful reading, positive comments, and constructive feedback.

We provide below a point-by-point response to the referees' comments (red = Referee's original comments, black = our response, blue = text from the revised manuscript, verbatim).

Referee 1

We thank the Referee for their thorough reading of the manuscript and are grateful for them positively recommending our paper for publication and for affirming our paper to be "interesting", the technique "clearly demonstrated", and that we have provided a "new tool for quantum simulation and quantum computation". We are happy to address the excellent points they raised to help us improve the manuscript for the broader audience, listed below:

1) Concerning Extended Rydberg blockade range.

a) I believe that working with static, small Rabi frequency, instead of modulation, can in principle, give the same result. It is nice that the authors discuss the merit of FFM scheme over the static weak excitation scheme. Note that, in this discussion, the authors should compare the two schemes on the SAME Rabi frequencies, meaning that in the FFM scheme the authors should use the suppressed Rabi frequency $J_0 \times \Omega$, instead of bare Ω as in the comparison in SI IV.

The Referee is right in pointing out that working with a small static Rabi frequency $\Omega_s = \Omega J_0(\alpha)$ allows one to achieve the same absolute Rydberg blockade radius. Comparing between FFM and the static weak scheme, the former can be technically simpler to realize a large dynamic range of control. The steps required to control a range of Rabi frequencies for each scheme are laid out more explicitly in a new paragraph of section IV of the supplementary information (SI):

We note that the same effective extension of the Rydberg blockade range can be achieved in the static scheme with a lower Rabi frequency Ω_s , where $\Omega_s = \Omega J_0(\alpha)$. However, dynamic control of the Rydberg blockade range through FFM can potentially be technically simpler than that in the static scheme, especially where the atoms require two-color Rydberg excitation. In the latter case, accurately controlling a range of Rabi frequencies would require calibrating out the change in differential light shift that accompanies the nominal change in Rydberg laser intensities. Achieving the desired Rydberg laser intensities in turn requires a careful characterization of the AOM diffraction efficiency, which depends on both its RF drive frequency and amplitude. In contrast, FFM only requires the latter characterization of the AOM to realize a drive with minimal residual amplitude modulation. Further, the control of Rabi frequencies over a large range (0 to Ω) can already be achieved with a fairly small range of modulation indices (e.g. $\alpha = 2.4$ to 0).

1b) In addition, smaller effective Rabi frequency requires the longer time for Rydberg two-qubit gate, leading to larger infidelity of the two-qubit gate, due to the finite lifetime of Rydberg state. Can the authors discuss this point?

We thank the Referee for raising this important point. We address this in two parts.

First, in the main manuscript, where we mention how the gain in entanglement distance can come at a cost of longer gate times, we amend the text to include the finite Rydberg state lifetime as a consideration:

Where the coherence of the Rydberg excitation laser and the finite lifetime of the Rydberg state are no longer dominant constraints, the entangled state fidelity can be optimized by choosing alpha in exchange for longer gate times.

Second, we have edited SI section IV to discuss the intrinsic limit on the two-qubit gate fidelity, which is set by both the Rydberg lifetime and the van der Waals interaction strength. We note here that experimental imperfections can lead to deviations from the intrinsic limit, thereby giving rise to a dependence of the two-qubit gate fidelity on the gate times. Nevertheless, since the intrinsic limit is already compromised at longer entanglement distances due to the reduced interaction strength, we focus instead on the intrinsic limit. The last part of SI IV now reads:

We note, however, that as long as the interaction strength V is decreased, the two-qubit gate fidelity will be compromised, as shown by the following expression for the intrinsic two-qubit gate error¹⁰ E_{min} :

$$E_{min} = \frac{3(7\pi)^{2/3}}{8} \frac{1}{(V\tau_R)^{2/3}}$$

where τ_R is the Rydberg state lifetime.

2) Concerning Protection of entangled state against dephasing

Intuitive explanation on the stronger protection of the entangled state via FFM over the free evolution is only briefly provided as arising from a high-frequency interruption of the dephasing process. I think this is not enough and more detailed discussion is helpful for broader range of readers. In particular, I am wondering if this protection is related with a quantum Zeno effect.

We thank the Referee for motivating us to clarify the origin of the coherence protection. In the main manuscript, we explain that the dephasing interruption comes from the residual weak coupling between the $|W\rangle$ state and $|ee\rangle$, where the $|ee\rangle$ is insensitive to drifts in relative phase shifts:

Intuitively, the protection arises from a high-frequency interruption of the dephasing process. More specifically, although we null out the Rabi coupling between $|gg\rangle$ and $|W\rangle$ with FFM, the off-resonant coupling between $|W\rangle$ and $|ee\rangle$ remains. Since the $|ee\rangle$ state is robust against drifts in relative phase shifts between the atoms, the Doppler dephasing is periodically interrupted whenever the system evolves to pick up the small $|ee\rangle$ admixture. The fast residual oscillation is washed out in a time average, giving rise to a net protection of the $|W\rangle$ state.

Referee 1 also mentioned the quantum Zeno effect, which is a nice analogy to the dephasing interruption. However, the two are not the same because no measurement is carried out when the dephasing is interrupted.

3) Concerning Enhanced Rydberg anti-blockade dynamics

I believe that the Rydberg excitation with static, two frequency components, one resonant with the gg to W and another with W to ee , instead of modulation, can in principle, gives the same result. Actually, the authors used this later scheme for the measurements described in Supplementary Information II. It is nice that the authors discuss the merit of FFM scheme over the static, two frequency components, excitation scheme.

Since realizing the STIRAP scheme that the authors discussed is rather straightforward for the authors, I am wondering if there is any technical issue. If so, it is nice to discuss this point.

We agree with the Referee that a static, bichromatic drive can give rise to population in the $|ee\rangle$ state. That said, before comparing FFM to the static, bichromatic scheme, we first clarify that SI section II uses instead two consecutive monochromatic pulses to drive atoms to the $|ee\rangle$ state:

first, the two atoms are excited from $|gg\rangle$ to $|W\rangle$ with a resonant monochromatic π -pulse ($\delta = 0$); subsequently, $|W\rangle$ is excited to $|ee\rangle$ with a second monochromatic pulse of variable detuning δ' .

Compared to pulses, STIRAP is preferred for its robustness against timing jitter. Indeed, both FFM and the static, bichromatic scheme can perform STIRAP, but the bichromatic scheme would require individual control over at least 3 lasers in this case. Conversely, FFM does not require additional hardware beyond what is used for the conventional monochromatic scheme and is thus simpler. This is discussed at the end of SI section VIII:

We have pointed out in the main text that it is not possible to populate the $|ee\rangle$ state for two closely spaced atoms via STIRAP with the conventional monochromatic static excitation scheme. That said, one can adopt a version of STIRAP with two static frequency components. In this case, where each single-atom Rydberg excitation requires two colors, the bichromatic STIRAP scheme would require at least three lasers with individual control over each laser's detuning and intensity. In contrast, FFM is simpler to implement as it does not require additional optomechanical components beyond what is needed for a monochromatic Rydberg drive.

To avoid confusion in the main text, where we state that steady-state $|ee\rangle$ population cannot be achieved in the static scheme for closely-spaced atoms, we clarify that we mean the conventional scheme:

- Paragraph 4: Such a strongly-interacting state cannot be otherwise attained in the steady state with a **conventional** static drive...
- Page 5: ... yet the steady-state population of multiply-excited Rydberg states cannot be achieved by applying STIRAP with the **conventional** static excitation scheme.

Finally, we explain in SI section VIII the technical reason for not executing the STIRAP scheme:

Nevertheless, since the time required to do the STIRAP transfer is $4 \mu\text{s}$ and is comparable to our present coherence times, we did not attempt a demonstration of our proposed STIRAP scheme here.

4) Concerning Fig. 4

Fig.4c): I cannot find any explanation for red, green, and blue shading. I can imagine they are MC simulations, but explicit explanations are helpful (although in SI, there are related sentences.)

Fig. 4d): Similarly, I cannot find any explanation for grey shading.

We have amended the captions for Fig 4c and Fig 4d, which now read:

c Measured population dynamics (top) ρ_{ee} , (middle) $\rho_{eg} + \rho_{ge}$, and (bottom) ρ_{gg} for $\alpha = 1.4$, which are in good agreement with Monte Carlo simulations represented by shaded curves (in same color as data points).

d ... (Gray line) With ground state cooling and enhanced coherence times of $74 \mu\text{s}$, the mean steady-state population becomes $\rho_{ee} = 0.85$, with one standard deviation of the Monte Carlo sample distribution depicted as the half width of the light gray band.

5) Concerning Methods

At the beginning, the authors state "...two magic-wavelength tweezers ...". I am a little bit confused about this wording. Does this "two" means "two sites" ? How many two-sites are used in the experiment ?

Only one pair of tweezer sites are used in the experiment. For clarity, the first line in the Methods section now reads:

Our experiment begins with loading individual ^{23}Na atoms into a pair of magic-wavelength tweezer sites from a precooled reservoir.

6) Concerning Supplementary Information I

In the evaluation of false negative error, the authors used the BBR-limited lifetime for the Rydberg state. Since the conversion to the ground state should occur for this error and the BBR induces the transfer to other Rydberg states, this BBR-limited lifetime does not seem to provide the relevant time constant.

We are grateful to the Referee for pointing this out and have corrected SI section I to use the natural lifetime (260 μs) rather than the BBR-limited one. We have also updated the manuscript figures to reflect the new false-negative errors in our plots of the Monte-Carlo simulations. SI Section I now reads:

... Given the natural lifetime of 260 μs for the $59\text{S}_{1/2}$ Rydberg state², we expect a false negative error of 0.02. ... Therefore the total false negative error is 0.03.

Correspondingly, on page 2 of the main manuscript, we report the false negative error to be 3%.

7) Concerning Supplementary Information II

Can the authors compare their method with much simpler one from the imaging magnification? The presented method looks nice, but could be model-dependent?

The Referee is correct in that our distance calibration method is model-dependent; for instance, it relies on knowledge of the value for C_6 . We obtained our value for C_6 from the Alkali.ne Rydberg Calculator python package, now cited as Ref [4]. We further assume an overall distance-independent frequency offset δ_u arising from an imperfection in determining the unshifted resonance. The fit function used is therefore

$$\delta'(f_r) = \frac{C_6}{(\kappa f_r)^6} + \delta_u$$

where κ and δ_u are fit parameters. Nevertheless, the presented method is preferred over using the distance calibration from the tweezer magnification, as our method allows us to directly measure the interaction strength, which is the quantity of interest in the Hamiltonian.

In any case, the distance calibration obtained from our method is in reasonable agreement with the expected value from the tweezer magnification, as we now explain in the last part of SI section II:

The fit distance calibration of 0.780(2) $\mu\text{m}/\text{MHz}$ is consistent with the expected value (0.783 $\mu\text{m}/\text{MHz}$) calculated from the acousto-optic deflector specifications, objective focal length, and tweezer telescope magnification.

8) Concerning Supplementary Information Extended Data Fig. 5

I am a little bit confused. In the caption, are the green and grey swapped?

In addition, it may be helpful to explicitly show that the displayed curves are for the static drive scheme (shown possibly in the figure).

We thank the observant Referee for catching this error and have amended the caption for Extended Data Fig 5 to swap “gray” with “green”. For clarity, we have changed the horizontal lines reflecting the

FFM entangled state fidelities to dot-dashed lines. Finally, we have indicated in the figure legend that the solid curves are for the static drive.

Referee 2

We thank the Referee for their strong, positive summary of our work, including their enthusiastic comments on how we have “**successfully demonstrated various exciting phenomena, which is an excellent achievement**”, and that the “**results are very solid, and also the paper is well written**”. We appreciate their recommendation for the paper to be published and for their constructive feedback, which we have now incorporated into the revised and improved manuscript.

1) The introduction's second paragraph states, “...to prevent substantial entanglement infidelity due to double Rydberg excitations”. In a general context, the statement is not accurate. I feel, for instance, one may need a double excitation population to realize a maximally entangled state like $|gg\rangle + |ee\rangle$. So, a better reformulation would be suitable.

The Referee has raised a good point about considering other Bell states. To avoid confusion, we have reformulated the sentence to specify that we are referring to the Rydberg blockade error. The second paragraph of the introduction now reads:

... to prevent substantial entanglement infidelity due to **blockade error**.

2) In the paragraph above eq.2, the state $|ge\rangle + e^{i\phi}|eg\rangle$, does the ϕ can take any values, for instance, π ? for which one creates the singlet state $|ge\rangle - |eg\rangle$? Or ϕ is restricted to some values?

We clarify in the revised Methods section that the atom array axis is perpendicular to the Rydberg excitation lasers, therefore

the relative phase ϕ in the expression for $|W\rangle$ is close to zero.

As a point of interest, we note here that if the atom array axis is arranged to lie parallel to the Rydberg lasers and where the array spacing can be tuned, ϕ can take on a range of values including π and is simply set by the initial positions of the atoms. In other words, an initial random position difference Δx leads to a unitary rotation of the Bell states $\{|+\rangle, |-\rangle\}$ into the $|W\rangle$ and $|D\rangle$ states:

$$\begin{aligned} |+\rangle &\propto (|ge\rangle + |eg\rangle) \rightarrow |W\rangle \propto (|ge\rangle + e^{ik\Delta x}|eg\rangle) \\ |-\rangle &\propto (|ge\rangle - |eg\rangle) \rightarrow |D\rangle \propto (|ge\rangle - e^{ik\Delta x}|eg\rangle) \end{aligned}$$

Regardless of the spatial arrangement, we reiterate in the revised SI section V that:

the relative phase $k(x_2^{(i)} - x_1^{(i)})$ is nominally fixed for each experiment trial.

Since they are fixed for each trial, the above unitary rotation does not affect measurements or fundamental dynamics. This point is also explained in SI reference 6.

3) The discussions on the second page, starting from the paragraph, “We point out that FFM is distinct from the Floquet engineering ...” and the paragraph following that, are best suited for the introduction.

As per the Referee’s suggestion, we have moved the one of the two paragraphs, “A versatile approach, FFM has been used...” to the introduction. For the paragraph starting with “We point out that FFM is distinct from the Floquet engineering...”, we opted to keep it in its same position on the second page

because it explains that “FFM directly transforms the couplings”, which we feel is best appreciated after Equations (2) and (3) have been discussed.

4) The references Refs.[25, 26] above eqs.(2) and (3) discuss theoretically some of the main ideas in this paper, such as blockade at small interactions, antiblockade at large interactions, and Bell-states freezing using FFM. That needs to be appropriately credited in the draft and best suited to be included in the introduction. As the present paper demonstrates the experimental results, it has its merit.

We sincerely thank the Referee for raising this important point and have amended the manuscript to ensure that Refs [25, 26] are appropriately credited. We do this in four places -

1. In the introduction, we amend the text such that it now reads:
... First, we demonstrate that atoms can be entangled outside the traditional blockade radius, **as predicted by Refs** ^{25, 26}, thereby significantly increasing the useful range of the Rydberg interaction. Furthermore, we show how FFM can protect a two-atom entangled state against Doppler dephasing, which is the typical mechanism limiting entangled-state coherence in a Rydberg atom array. Finally, we propose a robust transfer of closely-spaced atoms from the ground state into an anti-blockaded state....
2. The caption for Fig 1 (which we refer to in the introduction) is changed to include:
The counter-intuitive regime of interactions can be exploited with Floquet frequency modulation, where atoms outside the static blockade radius experience Rydberg blockade and vice versa, **as predicted by Refs** ^{25, 26}.
3. On page 3, where we discuss protection of the entangled state against dephasing, we cite Ref 26:
The $|W\rangle$ state can be dynamically stabilized²⁶ under the same conditions that give rise to population trapping ($J_0(\alpha) = 0$).
4. On page 4, where we discuss enhanced Rydberg anti-blockade dynamics, we insert citations to Refs 25 and 26:
On the other hand, FFM provides a convenient handle to access the $|ee\rangle$ state from the $|gg\rangle$ state by fulfilling both resonance conditions^{25,26} described in Eqs. (2) and (3) (Fig. 4a).

5) It would be useful if authors can provide a reference for the statement “the protection arises from a high frequency interruption of the dephasing process”. As it is very important point in the scenario of the experiment. Or add more insights or explanations into it, how the FFM can suppress the decoherence.

We thank the Referee for motivating us to improve the quality of our discussion on the entangled state coherence protection. To the best of our knowledge, the protection against Doppler dephasing afforded by FFM is a novel feature of this work. We recognize that it warrants a more detailed explanation, so we have added some text on page 4 of the manuscript, attributing the high-frequency interruption to the fast residual oscillation between $|W\rangle$ and the admixture of $|W\rangle$ with a small $|ee\rangle$ component.

For a full explanation, we refer Referee 2 to our response to Referee 1, point 2.

Referee 3

We thank the Referee for their recognition of how our work is “novel and will be of interest to the growing community of Rydberg-array-based quantum physics” and for their positive comments on how “the paper is soundly written in general, and the claims are well supported with experimental and numerical results”. All the Referee’s expert comments are much appreciated, and we address them below.

1) The authors demonstrate and discuss the application of FFM in entanglement generation, but can it also be used in a Rydberg quantum logic gate? If yes, what are the Pros and Cons vs the conventional blockade-based Rydberg gate scheme?

Yes, FFM can be used in a Rydberg quantum logic gate and is compatible with the conventional blockade-based Rydberg gate scheme (e.g. the Levine-Pichler protocol) between atoms that encode the qubit states in the ground hyperfine levels. We explain this in detail in the revised version of SI section VI, which now includes an additional figure (Extended Data Fig. 6) outlining the FFM-modified Levine-Pichler protocol. SI section VI now reads:

While this work focuses on using the ground state $|g\rangle$ and Rydberg state $|r\rangle$ as the two qubit states, one can apply the FFM scheme to entangle atoms that encode their qubit states in two (e.g. hyperfine) ground states, denoted here as $|0\rangle$ and $|1\rangle$. For instance, FFM is readily compatible with the Levine-Pichler protocol⁸ that has been used to implement a two-qubit controlled-phase gate in the Rydberg blockade regime⁹. The conventional Levine-Pichler protocol uses two global Rydberg pulses of duration τ , sandwiched by a phase jump, to drive partial or full Rabi oscillations on two-qubit states with at least one qubit in $|1\rangle$. Where an extended Rydberg blockade range is desired, the global Rydberg pulses can be applied with a frequency-modulated detuning and with a user-defined modulation index α (Extended Data Fig. 6).

For a comparison against the fully static conventional blockade-based Rydberg gate scheme, we refer Referee 3 to our response to Referee 1 points 1a and 1b.

2) The finite anti-blockade population is attributed to the interaction disorder from the position spread of the atoms with 1 uK temperature, and the authors propose to mitigate this effect with further cooling. It seems to me that, the severe trap-depth ramping from 1.3 mK to 7 uK would largely increase the initial atomic wavepacket distribution. So, if the interaction disorder is mainly caused by initial position spread (instead of initial atomic velocity), then maybe it can be mitigated just by going easy on the trap-depth ramping?

The Referee is absolutely right in pointing out that we could have gone easy on the trap-depth ramping to minimize the interaction disorder at short times (e.g. $\leq 1 \mu\text{s}$ for the maximum anti-blockade population in Fig 4c). We acknowledge this point in the revised SI section IV:

In Fig. 4 of the main text, we note that the $|ee\rangle$ population at short times (Fig. 4c) can be boosted by using a less aggressive ramp of the trap depth, since the interaction disorder there mainly comes from the initial position spread. Where we consider the anti-blockade dynamics under ground-state cooling, we assume a final motional state occupation...

For consistency, we amend page 5 of the main manuscript to read:

This problem can be mitigated by optimal cooling techniques, such as motional ground-state cooling⁴⁵, and by increasing the atom-laser coherence...

REVIEWERS' COMMENTS

Reviewer #1 (Remarks to the Author):

I found that the authors satisfactorily revised the manuscript for my claim. Other revisions for the comments by the other Reviewers are also appropriate and satisfactory. Therefore, now I recommend the publication of this manuscript in Nature Communication.

Reviewer #3 (Remarks to the Author):

The authors have adequately addressed the comments from all referees and modified the manuscript accordingly. I think the paper can be published as it is.

Response to the referees

We sincerely thank the referees for their positive comments on how “the authors have adequately addressed the comments from all referees” and for their recommendations for the manuscript to be published.